# Supervised Myoelectrical Hand Gesture Recognition in Post-Acute Stroke Patients with Upper Limb Paresis on Affected and Non-Affected Sides

**DOI:** 10.3390/s22228733

**Published:** 2022-11-11

**Authors:** Alexey Anastasiev, Hideki Kadone, Aiki Marushima, Hiroki Watanabe, Alexander Zaboronok, Shinya Watanabe, Akira Matsumura, Kenji Suzuki, Yuji Matsumaru, Eiichi Ishikawa

**Affiliations:** 1Department of Neurosurgery, Graduate School of Comprehensive Human Sciences, University of Tsukuba, 1-1-1 Tennodai, Tsukuba 305-8575, Ibaraki, Japan; 2Center for Cybernics Research, Faculty of Medicine, University of Tsukuba, 1-1-1 Tennodai, Tsukuba 305-8573, Ibaraki, Japan; 3Department of Neurosurgery, Faculty of Medicine, University of Tsukuba, 1-1-1 Tennodai, Tsukuba 305-8575, Ibaraki, Japan; 4Ibaraki Prefectural University of Health Sciences, 4669-2 Amicho, Inashiki 300-0394, Ibaraki, Japan; 5Center for Cybernics Research, Artificial Intelligence Laboratory, Faculty of Engineering Information and Systems, University of Tsukuba, 1-1-1 Tennodai, Tsukuba 305-8573, Ibaraki, Japan

**Keywords:** stroke rehabilitation, hand gesture recognition, surface electromyography, supervised learning, support vector machine

## Abstract

In clinical practice, acute post-stroke paresis of the extremities fundamentally complicates timely rehabilitation of motor functions; however, recently, residual and distorted musculoskeletal signals have been used to initiate feedback-driven solutions for establishing motor rehabilitation. Here, we investigate the possibilities of basic hand gesture recognition in acute stroke patients with hand paresis using a novel, acute stroke, four-component multidomain feature set (ASF-4) with feature vector weight additions (ASF-14NP, ASF-24P) and supervised learning algorithms trained only by surface electromyography (sEMG). A total of 19 (65.9 ± 12.4 years old; 12 men, seven women) acute stroke survivors (12.4 ± 6.3 days since onset) with hand paresis (Brunnstrom stage 4 ± 1/4 ± 1, SIAS 3 ± 1/3 ± 2, FMA-UE 40 ± 20) performed 10 repetitive hand movements reflecting basic activities of daily living (ADLs): rest, fist, pinch, wrist flexion, wrist extension, finger spread, and thumb up. Signals were recorded using an eight-channel, portable sEMG device with electrode placement on the forearms and thenar areas of both limbs (four sensors on each extremity). Using data preprocessing, semi-automatic segmentation, and a set of extracted feature vectors, support vector machine (SVM), linear discriminant analysis (LDA), and k-nearest neighbors (k-NN) classifiers for statistical comparison and validity (paired *t*-tests, *p*-value < 0.05), we were able to discriminate myoelectrical patterns for each gesture on both paretic and non-paretic sides. Despite any post-stroke conditions, the evaluated total accuracy rate by the 10-fold cross-validation using SVM among four-, five-, six-, and seven-gesture models were 96.62%, 94.20%, 94.45%, and 95.57% for non-paretic and 90.37%, 88.48%, 88.60%, and 89.75% for paretic limbs, respectively. LDA had competitive results using PCA whereas k-NN was a less efficient classifier in gesture prediction. Thus, we demonstrate partial efficacy of the combination of sEMG and supervised learning for upper-limb rehabilitation procedures for early acute stroke motor recovery and various treatment applications.

## 1. Introduction

Post-stroke motor deficits directly influence post-stroke recovery due to reduced self-care ability and participation in social life, leading to poor outcomes in one-third of survivors within 5 years after onset [1,2,3]. The slow and often poor recovery process after both ischemic and hemorrhagic cerebrovascular incidents, thus, places enormous financial burdens on society and individuals [2]. Stroke recovery is complicated by lesions, the deposition of which is unique for each patient, and common rehabilitation procedures cannot often be applied under such diverse requirements with expectations of effectiveness [1,4,5,6,7,8]. However, a core rehabilitation tactic that is broadly applicable in stroke recovery relies on motor learning, which is the keystone of neuroplasticity for further recovery and adaptation of preserved limb motor skills [9,10].

Healthcare fields are currently undergoing explosive growth in fields related to individualized and user-centric rehabilitation for neurological motor impairments that promote motor learning [11], such as semi-stationary type robotic manipulators [12,13], cyborg-type suits [14,15], portable orthoses [16], and medical extended reality [17]. These emerging, noninvasive medical applications rely chiefly on skin surface electromyography (sEMG) [11,18], which measures bioelectrical potential on the affected limb’s skin and further amplifies signal intensity as a mechanical output to achieve feedback or to drive simulated motor activity within a virtual environment during rehabilitation procedures [19]. Particularly for stroke impairment recovery, animal and clinical data indicate a definite advantage of feedback-driven interactions for motor learning in enriched environments [17,20]. Such enriched environments for post-acute stroke patients must be tailored with specific afferent feedback standards since musculoskeletal and cognitive behavior dynamically change after onset, especially in the first weeks after lesions manifest [19,20,21,22]. However, the myoelectrical behavior of early-stage stroke survivors is fractional and varies based on afflictions such as spastic co-contraction, previous stroke lesions, behavioral impairments, and even post-stroke depression [3,4,5,6,7].

Paretic hand bioelectrical signal evaluation as a key domain for feedback-driven solutions can be partially reflected in the field of limb prosthetics in which the priority is artificial limb control by tracking the remaining atrophic muscle signals [23,24,25]. Subsequently, sEMG-driven prosthetic control workflows, analogous to paretic myoelectrical pattern recognition reported by Hudgins [26], expanded the scope of bioelectrical signal processing into related medical areas [11,13]. A substantial breakthrough in sEMG signal processing for neurorehabilitation has been achieved with the assistance of wearable electronics and other portable signal detectors, such as the Myo Armband wearable bracelet manufactured by Thalmic Labs [16,27,28,29]. Technically, sEMG decoding for pattern recognition within the upper extremities and hand gestures requires computational processing that has recently centered on machine learning; such advancements in this field are expected to synergistically increase the resolution of signal capture and processing for rehabilitation use as individual patient requirements can be learned by the software [28]. While stroke-related musculoskeletal signal pattern recognition is highly variable, the initial procedure steps remain constant: data gathering, labeling and segmentation, preprocessing, subsequent normalization, muscle signal feature extraction, feature vector concatenation, feature selection, further gesture synergy classification, and statistical postprocessing. In practice, sEMG electrodes are usually attached to the affected forearm muscles (both flexors and extensors) to aid in synthesis of signal processing [29,30,31]. Depending on the grade of paresis and study design, the number of electrodes may vary, from two-channel up to high-density systems, on the entire surface of the forearm [32,33,34]. In real-time applications [35,36,37], the frequency and number of channels determine the structure of the raw sEMG signal, allowing for the extraction of valid signal features for pattern recognition processing by adaptive and intelligent machine systems.

Recently, valid feature selection for gesture recognition in paretic hands is being extensively reported due to the capacity for wide-ranging application to diverse methods [27,38]. Among the top issues are the extracted feature domain, the impact of dimensional concatenation of various feature domains on performance, appropriate wavelet transforms, and types of classifiers [31,39]. As an example, some research groups rely on the recognition of hand gestures in healthy individuals, in which time-domain features are mainly represented as the base for subsequent experiments. An illustration of this is the Hudgins feature set and its modifications [34,35]. However, there are studies indicating that time-domain features of sEMG are susceptible to muscle noise and artefacts while additional data indicate that frequency-domain and time–frequency-domain features are less redundant with regard to muscle spasticity for managing nonlinear signal parameters [33]. In summary, the major consensus in the post-stroke myoelectric pattern recognition field states that the support vector machine (SVM) classifier has a dominant role in valid paretic sEMG pattern recognition [40] and can sometimes be compatible with even artificial neural network (ANN) algorithms.

Overall, previous reports on sEMG-driven gesture recognition for stroke were narrowly focused on the chronic stage, although post-acute stage applications remain challenging [29,30,31,32,33,34,35,41]. For each signal processing step, the acute stroke data may compromise valid real-time predictions and remain unsolved, while heterogeneous variation of central motor impairments results in non-static distortions and further degradation of muscle activity in the affected limb. These impairments are reflected in technical designs that might require specific electrode placement on the affected extremity, spasticity resilience feature vectors, and filtering of dimensional overweighting of parameters which could overwhelm classification and prediction capacities [42,43].

However, the acute stage of stroke requires specific attention since recovery plateaus and remains constant within 1 month after onset [4]. Additionally, rehabilitation efforts in the 3–6 month post-stroke period may benefit from enhanced neural plasticity and focused efforts with sEMG-driven treatment platforms would be poised to take full advantage of this with their ease of use and adaptability [21]. Since nearly 80% of survivors have interim motor impairments (with half of such patients retaining deficits in hand and finger movement functionality), it is essential to improve rehabilitation and adjust the motor training individually, especially for the hand [5,8]. ADL-oriented motor functioning dynamic analysis, derived from the patient’s individual characteristics, may expand the understanding of rehabilitative recovery tactics and help occupational therapists tailor training and design exercises to improve focused outcomes [22,44]. Accordingly, as a hypothesis, we investigated the viability of basic myoelectrical hand gesture recognition in patients to identify the most effective design settings suitable for cross-validation, aiming to achieve user-independent application for further fine motor recovery and timely recovery monitoring diagnostics in the critical post-stroke time period.

## 2. Materials and Methods

### 2.1. Participants

We conducted an observational study of 59 patients, among which 19 patients (65.9 ± 12.4 years old; 12 men, seven women) had an acute stroke with brachial monoplegia. The candidate pool included hospitalized patients admitted to the University of Tsukuba Hospital within the post-acute stroke period (or for cerebrovascular incidents within the period of hospitalization) from December 2020 to June 2021.

Written, informed consent was obtained for every participant. Eligible hospitalized patients who passed screening tests were assessed during routine procedures with an occupational therapist. Organizing and collecting data took 1 h, on average, and did not interfere with scheduled treatment procedures. Study regulations were approved by the University of Tsukuba Hospital Ethics Committee (approval number: R02-204). Research protocols and design were implemented according to the Declaration of Helsinki (or equivalent) and the University Guidelines for Clinical Trials.

Physical examinations were evaluated with the following stroke upper-limb motor impairment scales: Brunnstrom stage (BS; 1 to 6 points), which measures the state of post-stroke movement recovery; Fugl–Mayer assessment for upper extremity (FMA-UE; 0 to 66 points), which evaluates limb functioning; the single-task various function evaluation Stroke Impairment Assessment Set (SIAS; 0 to 5 points); the Modified Ashworth Scale (MAS; 0 to 4 points), where lower scores represent normal muscle behavior and higher scores indicate the development of post-stroke spasticity [45,46,47].

### 2.2. sEMG Recording Setup and Observational Experiments

To establish the technical aspects of the procedure, we used removable, portable foam hydrogel Ag/AgCl sEMG sensors using a wireless eight-channel Biosignal Plux device (PLUX Wireless Biosignals S.A., Lisbon, Portugal) certified for biomedical research. The real-time recording mode was set to 16 bit analog-to-digital (ADC) converter resolution and 1000 Hz sampling rate.

Participant hands and forearms were prepared with alcohol wipes to minimize possible artefacts and deviations during the sEMG recording. Before electrode attachment, forearm muscles were palpated by specialists. Five total sensors were placed according to the functional upper-limb anatomy, targeting muscle bellies and forearm anterior/posterior locations (sensors were placed symmetrically on non-affected and affected arms). Sensors were placed on the projection of the flexor carpi radialis muscle, the flexor carpi ulnaris muscle area, the abductor pollicis brevis muscle in the thenar area, and the extensor digitorum communis muscle, plus a grounding sensor that was placed on the lateral elbow epicondyle. We aimed to preserve 2 cm of distance between the electrodes [48]; however, due to the specific individual constitution of some patients and medical conditions (e.g., forearm arteriovenous fistula for hemodialysis), those parameters were not achieved in every case.

The procedure involved a one-time sampling of surface electromyography from patients in a controlled environment to minimize possible biases and monitor participant conditions. In the research design (Figure 1), patients in moderate or stable condition repetitively performed the following hand gestures 10 times each at the same interval from the idle hand posture: fist, pinch, wrist flexion, wrist extension, finger spread, and thumb up. Those clinically relevant gestures represent the basic components of ADLs which would be simultaneously and gradually introduced during early-stage rehabilitation [15,49]. Such movements are classified as repetitive fine hand training (index finger and thumb movements) and strength exercises (muscle coactivation for multifaceted motor control such as grip).

### 2.3. Data Preparation

#### 2.3.1. Signal Preprocessing

During the experiments, post-stroke neurological impairments caused some of the patients to have difficulty in performing hand movements on the first attempt. Hence, non-segmented data had a portion of artefacts and signal gaps that were unfit for statistical evaluation and required additional preparation.

First, real-time bilateral electromyography data from both limbs were combined into two datasets with standardized dimensions. The initial raw signal was converted into a micro voltage (mV) format factor in accordance with the following transfer function that represents the device configuration (−1.5 mV, 1.5 mV) [50]:$$EMG\left(V\right)=\frac{\left(\frac{ADC}{2^{n}}-\frac{1}{2}\right)VCC}{G_{EMG}},$$
where the operating voltage *VCC* is equal to 3 V, sensor gain $G_{EMG}$ is 1000, *ADC* is equal to the sampled value from each channel, and *n* is the number of the channel bandwidths represented in bits.

Second, taking into consideration MAS scores and adjacent muscles on the forearm which could introduce movement artefacts, we applied high-pass (HPF) and low-pass frequency filters (LPF). According to related studies, filters defined by the International Society of Electrophysiology and Kinesiology (ISEK) standards of sEMG analog processing have initial ranges of 5 Hz cutoff for HPF and 500 Hz for LPF to ensure an optimal window signal range for upper-limb applications [42,51]. However, for managing muscle rigidity, we adapted the HPF/LPF ratio and added additional filtering preprocessing strategies [37,52]:A fourth-order Butterworth bandpass filter (BPF) ranging from 20 to 300 Hz;Hampel filtering for artefact reduction by identifying outliers deviating from an average of more than double the standard deviation in the neighboring 100 samples;Root-mean-square signal envelope and sEMG data normalization.

For our experiments, hand gesture recognition workflow included seven gesture labels (rest or idle state, fist, pinch, wrist flexion, wrist extension, finger spread, and thumb). The segmentation started from a manual selection of active segments that contained one label of a single gesture. Furthermore, except for the ‘rest’ label, we used a centered, moving average function for automatic peak detection to process temporal analysis for four channels simultaneously with the following settings:The maximum peak variable likelihood estimation was set to 8, cutting off the first and last gestures from each 10-gesture repeated set.Minimal temporal distances between the peaks were configured as 0.1 s.To avoid false variables, the estimation of peak selection was set at 0.25 percentile of the difference from neighboring signal peaks.For temporal standardization, signal boundaries (including the manually predefined ‘rest’ label) were determined by the window length of 30 ms for the inferior limit and 60 ms for the superior limit with regard to the peak.

To avoid peak misdetection, sEMG data were visualized and inspected (Figure 2); the usual misdetection rate for the non-affected side was around 5–7% and 12–15% for the affected side. Validation included manual window-length arrangement for non-detected variables. Finally, spliced datasets were demeaned and normalized.

#### 2.3.2. Feature Extraction

In terms of biophysical definitions, electromyography of the upper extremities can be expressed as a set of time series of myoelectrical static events; the multichannel summation of which can store specific patterns of the hand gestures in the form of vector features [29,30,31]. Following acute-stroke research design, we aimed to use corresponding real-time features for future applications and, thus, focused on preserving optimal accuracy rate precision and low-latency computational performance. Validated by iterations over the stroke dataset, we established the acute-stroke four-component multidomain feature set (ASF-4) that provides fast computational evaluation to accurately recognize simple gestures. ASF-4 features include a twofold feature component of the fourth and fifth temporal moments (TM4-5), median value of the energy of wavelet packet coefficient (MEWP), logarithmic cardinality (LCARD), and the Hilbert–Huang transform (HHT).

First, to emphasize the functionality of our ASF-4 set (Table 1), we used the high-order statistical baseline of TM4 and TM5 introduced by Saridis and Gootee for hand prosthetic control [23]. Temporal moments are used to measure the power spectrum of certain class estimations and use absolute value as a time-domain feature [27]. Second, we used the HHT time–frequency feature that contains empirical mode decomposition and the original Hilbert transform spectral function [53]. This given combination allows HHT to provide nonlinear and nonstationary analysis that has certain utility in muscle fatigue evaluation for biomedical applications [54]. Third, we revised the common application of cardinality [55] since it has a nondependent structure over the sEMG signal’s amplitude while, on the other hand, requiring threshold values for managing utilized units that affect the model precision. In our settings, we enlarged the potential of cardinality by turning it into a logarithmic form factor but setting a strict, low-number threshold for practicality. Finally, MEWP was added as feature with multiple wavelet coefficients of scaled signal length via wavelet decomposition, signal positioning, and frequency context [43,56]. The operation of this function was exclusively tuned to the median vector value in order to manage high-volumetric data in a usable form.

Considering the peculiarity of the myoelectric signature of post-stroke patients, we defined a diagram (Figure 3) for the initial selection of features that should reflect the specific movements of the hand and forearm: wavelet-oriented value shift of the time-scale function (MEWP and MEWT), the sEMG signal’s shape outliner, feature vector weight enhancing for supervised learning, and its adaptive approximation [40].

Initially, we applied our novel, stroke-oriented ASF-4 feature set as the base; however, in experiments, model performance was optimal for affected and non-affected extremities for three and four gestures, respectively. Further model enhancement met predictive pitfalls due to the fist (both flexor and extensor static muscle bursts) and thumb (included fist components posture) gestures.

To manage that issue, we added noise-resistant functions (both spatial and frequency characters) and our custom features to upscale multilabel class prediction (up to 6 gestures and rest state). Furthermore, we implemented nonlinear parameters to provide better accuracy for each gesture within the model [25,26,27,35,40,43,56,57,58,59,60,61,62]. In our settings, wavelet energy packet coefficients were stored as mean values for non-affected and median for affected sides. Amongst recent studies, that principle was partially disclosed by Phinyomark et al. in which the usage of median and mean values of frequency-domain features were able to reflect variegated muscle contractions and applied spatial force, which is vital for decoding the fine muscle tone of paretic forearms and hands [63]. This shapeshift helps to focus the abnormal muscle behavior and also provides complex signal decomposition across the signal length in non-affected extremities.

Keeping with the same approach, this tendency was also proposed for frequency-domain and time–frequency-domain features. In summary, aiming to control the level of model overfitting, we established two feature subsets of initial ASF-4 for non-paretic (ASF-14NP with 14 features) and paretic (ASF-24NP with 24 features) orientation.

ASF-14NP (Table 2) includes ASF-4, the logarithmic root mean square of second and third order (LRMSV2-3), the mean value of the absolute value of summation of exponent root (ASM) and absolute summation of square root (ASR), average amplitude change (AAC), Hjorth complexity (HPC), modified median frequency (MMDF), spectral median density (SMD), and flexor-to-extensor ratio value (FER-4).

ASF-24P (Table 3) includes ASF-4 (but MEWP tuned to the mean energy of wavelet coefficients MEWT), modified mean absolute values (MMAV2,5), simple square integral (SSI), kurtosis (KT), LRMSV2, ASM, ASR, standard deviation (SD), maximum fractal length (MFL) and wavelength (WL), multiple Hamming windows (MHW), third-order autoregressive and predictive linear coefficients (AR3, LPC3), modified amplitude spectrum (MASP), modified mean frequency (MMNF), spectral mean density (SND), short-time Fourier transform (STFT), Stockwell transform (SWT), and Higuchi fractal dimensions (HFD).

#### 2.3.3. Feature Validation and Visualization

In our framework, as features were concatenated, we converted each participant’s 4-ch sEMG data to Euclidean distances (ED) and standard deviation (SD) as a measure of distance for statistical evaluation. The mathematical definition of the ED is the root of the mean squared difference between the pair-channel signal’s sEMG data, while SD is the measure of the variance [39].

As a separation index, the ratio of the ED to SD (RES) was used [43,56], having a rising tendency with the ED value but decreasing with the SD value. Before the classification, we visualized feature vectors by channels for both extracted feature sets for initial feature navigation (Appendix A).

#### 2.3.4. Feature Vector Dimensional Reduction

To investigate optimal performance settings, we applied principal component analysis (PCA) to tune concatenated feature vectors. PCA establishes dimensional reduction by removing similar linear subspace within the feature vector [56], thereby highlighting the most relevant and distinct contents from the preprocessed sEMG signal that might serve as a feature selection approach. Indeed, PCA serves as a frontline method in noise signal processing [14,38] even for four-channel sEMG applications [64]. However, in our settings, due to obvious domination of time-domain single per-channel dimension features in the non-paretic subsets and multidimensional components in the frequency domain in the paretic subsets, we decided to set specific dimension reduction cutout settings. We, thus, used a one-dimensional investigation step for vector parameters from 1 to 50, where an integer value of 50 reflects the near-to-half complete feature vector structure number (ASF-14NP and ASF-24P).

### 2.4. Machine Learning and Classification Algorithms

Normalized by average and standard deviation, the extracted feature vectors were the input for supervised classification. As previously mentioned, the nonparametric SVM classifier is a widely used algorithm in musculoskeletal pattern recognition [34,40]; however, for comparison and validity, we applied two more algorithms: linear discriminant analysis (LDA) [56,57] and k-nearest neighbors (k-NN) [24,29,57]. In distinction to ANN, these supervised algorithms have low-computational costs and deployability [40]. Since stroke motor impairment scores (BS, SIAS, and FMA-UE) vary, we fixed a roadmap in label classification starting from four to seven gesture labels (*GL*) using the rest gesture label as the baseline for evaluation of all possible combinations of hand gestures, where *GL4* has 20 combinations ($={\;}_{7-1}C_{3}$), *GL5* has 15 combinations ($={\;}_{7-1}C_{4}$), *GL6* has six combinations ($={\;}_{7-1}C_{5}$), and *GL7* has only one ($={\;}_{7-1}C_{6}$).

The idea is to set up gesture recognition environment from early recovery using the most distinct and generic gestures (e.g., wrist flexion versus wrist extension). To validate results, we applied 10-fold SVM cross-validation (with 100 iterations each) amid 19 participants (16 for training, three for testing), which resulted in a total accuracy rate of predicted and true gesture labels. This first classification was performed without PCA and subsequently with PCA to identify the impact of dimensional reduction on performance; we could, therefore, extend the usability of supervised learning in post-acute stroke paretic recognition.

### 2.5. Statistical Evaluation

In general, to evaluate the multiclass supervised classification of biosignal pattern recognition in stroke victims, we deployed statistical measures of accuracy (*Acc*) and *F*_1_-score (*F*_1_) which were derived from the confusing matrix output values of true positives (*TP*), true negatives (*TN*), false positives (*FP*), and false negatives (*FN*) [59]. Overall, these performance metrics indicate whether the given gesture was correctly classified or mislabeled and explore the specific ratio of prediction potential, as indicated below.
$$Acc=\frac{TP+TN}{TP+TN+FP+FN}.$$

The *F*_1_-score, as a harmonic mean of precision (positive predictive value) and recall (true positive rate), demonstrates the summarized accuracy of the method which is used in bioelectrical signal processing:$$F_{1}=\frac{2TP}{2TP+FP+FN}.$$

*Acc*, as the main performance metric obtained from SVM, LDA, and k-NN, was validated using canonical paired-sample *t*-tests (*p*-values were less than 0.05) in each gesture label group (*GL4–7*). Thus, in the statistical evaluation, we tested *Acc* and *F*_1_ in pairs amid classifiers for the non-affected side and affected side to reveal tendencies in prediction between the identic and different algorithms. The same procedures were performed for the PCA-dependent performance using certain best principal components (PCs) of each classifier to study dimensional reduction and statistical significance. For this testing, the best PC values were derived from the best harmonic mean scores.

Statistical evaluation was developed using custom scripts in the *MATLAB R2020a* (MathWorks, Natick, MA, USA) software environment.

## 3. Results

### 3.1. Patient Characteristics

In 19 total patients, 11 cases of cerebral infarction (CI; including from embolic, cardiogenic, and atherosclerotic sources or acute Moyamoya disease) and eight cases of intracranial hemorrhage (ICH; caused by thalamic, putamen, brainstem, or cerebral cortex bleeding) were studied. Participating candidates (Table 4) had the following average motor impairment scores: BS 4 ± 1/4 ± 1, SIAS 3 ± 1/3 ± 2, and FMA-UE 40 ± 20. According to the MAS scale, two patients had objective signs of progressing spasticity in early post-stroke stage, while eight out of 19 patients demonstrated general signs of upper-limb spasticity.

### 3.2. Accuracy Rates and Confusion Matrices of Paretic and Non-Affected Extremities 

In both paretic and non-paretic extremities, we revealed that the use of a fist gesture label led to a decrease in the predictiveness of the model by an average of 7.7% ± 2.4% (mean calculated within all classifiers). Appendix A illustrate the random confusion matrix (CM) charts of the gradation of *GLs* from four to seven on a non-paretic dataset (NP19), whereas Appendix A show the data from the paretic side (P19).

Results indicate that supervised learners were capable of distinguishing *GL4* and *GL5* combinations on both extremities with an average accuracy rate above 80% in most scenarios. On the affected side, we observed a trend that flexor-related gestures had the most label misclassification. Moreover, the model complexity of a single-class dispersion (ability to recognize certain gestures) was varied and organized in a specific fashion in every classifier.

### 3.3. PCA Dimensional Impact on Supervised Model Performance

To reveal the best settings in the PCA technique [56], we cross-validated every *GL* with each component from 1 to 50 having a one-dimensional step (Figure 4). This procedure required a computational load of over 100 h on a consumer-type computer (Windows 11 Pro, Intel^®^ Core™ i9-7900X 3.30 Hz, 128 GB RAM, 1 TB SSD). The dimensional impact on the CM charts is shown in Appendix A.

Here, we were able to visualize and conceptualize the PCA relationship with classification performance; for P19, SVM result metrics remained stable without the sharp accuracy spikes throughout the PCs, whereas the NP19 figure was heavily deformed when using small PCs numbers. LDA metrics had a similar shape but the P19 experienced an accuracy trend in which a smaller number of PCs had a greater recognition score for each *GLs*. For k-NN, figures turned out to be distorted in terms of accuracy where the optimal parameters were obtained from only low PCs values. As for harmonic mean or *F*_1_-score, in general, those metrics were correlated with accuracy, turned out to be sensitive in dimensional investigation, and had larger deviation parameters across *GLs*.

To conclude, the optimal PC parameters above 80–90% accuracy rate for classifiers were the following: 23 (NP19) and 35 (P19) for SVM (where extracted feature vector length is usually equal to half of the initial vector length); 8 (NP19) and 19 (P19) for LDA (this linear classifier tended to target one-third of the complete vector length both for ASF-14NP and ASF-24P); 5 (NP9) and 8 (P19) for k-NN (having only a fractional feature vector part comparison to complete feature vector length).

We also discovered that, according to the empirical results, a combination of LDA and PCA increased classifier performance by up to 12.2% in certain models (Figure 4, Appendix A). Notably, the removal of three candidates with severe upper-limb paresis (BS I, II) from the P19 shifted the paretic dimensional best-spot component only for SVM (from 23 to 19), thereby indicating tightening in the sensitivity of learner variables. For further comparison, the best PC values were derived from the best *F*_1_-scores.

### 3.4. Classifier Statistical Evaluation and Comparison Using Dimensional Shift

In the first instance, we gradually inspected the statistical validity of supervised learners and the dimensional shift for each *GLs* in order to specify the best PCA settings and understand the optimal feature vector weights.

For this task, we evaluated the model performance between the same classifiers for paretic and non-paretic sides without (Figure 5) and with the use of PCA (Figure 6). These figures demonstrate that, for mono-classifier comparison using PCA, statistical significance (*p* < 0.05) was not obtained in every gesture model, whereas only *GL6* demonstrated statistical validity among PC pairs.

Furthermore, we summarized the cross-dimensional comparison of *p*-values between the classifiers (Figure 7). In this multi-model visualization, each square state indicates a specific event: dark blue demonstrates a *p*-value less than 0.05 (statistically valid relationship); black indicates a nonsignificant comparison between classifiers with a *p*-value of more than 0.05 (statistically invalid relationship); light blue reveals an absence of statistical relationships since a paired *t*-test evaluation was not performed due to logical inconsistency (invalid comparison).

Lastly, classifiers with a max of mean *Acc* and *F*_1_ (SVM without and LDA with the PCA) were evaluated to validate pairs of highest interest for statistical validity (Appendix A). Among max performance scores, the *p*-value remained less than 0.05 for most of the pairs, except the *GL7* model in the paretic dataset, where significance for both *Acc* and *F_1_* was not observed without the use of specific PCA settings.

### 3.5. Summary of Hand Gesture Prediction Rates

To summarize the results, we visualized performance metrics in a single figure for each algorithm to inspect model complexity trendlines (Figure 8 and Figure 9). Labels on the x-axis indicate the mean accuracy rate of 42 possible combinations within the *GL* class, which include the presence of specific gesture. The relationship between *Acc* and *F*_1_ was correlated throughout gesture model complexity variations of all classifiers (Table 5), where the harmonic mean turned out to be sensitive due to a wide gradation of paresis and limited number of studied patients (with lower scores for ‘fist’ and higher for ‘rest’). 

For SVM, the performance remained similar (above 90% for NP19 and above 85% for P19) over an increasing number of gesture predictors. *F*_1_ remained stable for the non-paretic side but decreased for the paretic side starting from *GL5*. Contrary to SVM without using PCA, following the processing of dimensional, reduced feature vectors, performance dropped several percent in general and flipped the accuracy trendline, whereby *GL4* had higher scores and *GL7* the lowest.

For LDA, prediction capacities varied with a certain deviation between P19 (near to 80%) and NP19 (near to 85%). The linear parametric classifier had a tendency of increased accuracy rate related to the number of used gestures, with the best results at *GL7*, whereas the paretic side remained flat over the *GLs*. Using multidimensional layering of concatenated feature vectors, LDA upscaled predictive performance up to 9–10% for both datasets, targeting the paretic side as primary impact. *F*_1_ also increased for most of the gestures in each *GL*. As a consequence, the use of PCA elevated LDA performance and allowed for the outperformance of SVM for most myoelectric pattern recognition scenarios.

The k-NN algorithm, conversely, was less sensitive toward the *GLs*, thereby having close prediction results amid all gesture clusters. Moreover, k-NN indicated close accuracy to SVM amid both datasets and flattened the total predictive capacities between each gesture whereas the standard deviation of *Acc* and *F*_1_ between paretic and non-paretic sides were the lowest (Figure 8). Dimensional reduction did not have a strong effect on the accuracy of k-NN while the *F_1_* for each gesture model drastically decreased.

Referring to the limited number of patients and diverse grades of motor impairments within the sEMG dataset, the harmonic mean serves as a sensitive method for classifier evaluation [59], which can assess both the SVM capacities and the optimal number of gestures (Figure 10). These figures indicate that, for the paretic side, the number of predictors (i.e., gestures) was critical for successful sEMG decoding and only partially decreased the recognition potential when using PCA in four and five gesture labels. The non-affected side, in contrast, was generally not limited by the number of gesture predictors in the model.

In summary, despite abnormal forearm and hand muscle behavior seen in the medical state of post-stroke survivors (Table 4), the best evaluated total accuracy rates using SVM among four, five, six, and seven hand gesture labels were 94.80%, 94.19%, 94.13%, and 94.73% for non-paretic and 88.71%, 88.48%, 88.60%, and 89.75% for paretic extremities. Throughout the experiments, SVM showed better scores in terms of *Acc* and *F_1_*, as well as performance stability, compared to LDA and k-NN for both extremities. LDA had a reduced performance compared to SVM for both paretic and non-affected extremities without additional dimensional settings (Table 5). Lastly, k-NN was able to preserve sufficient accuracy scores for the paretic and non-paretic sides with and without PCA. On the other hand, by having only a fractional feature vector for classification using k-NN, the dimensional shift did not change the single-gesture class prediction in *GLs*.

## 4. Discussion

The first goal of the study was to validate the hypothesis of hand gesture recognition in patients with post-stroke paresis (on affected and non-affected sides) at the early acute stage (within a month) using only a simple, four-channel sEMG system.

To start, according to the results (Figure 8 and Figure 9), non-paretic and paretic myoelectrical inputs vary, a fact also described in gesture recognition research for chronic stroke patients with forearm muscle atrophy [33,41]. It is worth noting that, within the prism of myoelectrical pattern decoding, the non-affected side cannot be treated the same as the healthy side. Therefore, gesture recognition studies mainly conducted among healthy subjects or amputee individuals are insufficient sources and references for post-stroke paretic muscle signal processing. First, stroke-related incidents are usually within a specific age group, which differs in terms of biomechanical upper-limb characteristics and data acquisition compared to healthy individuals [9,11,33]. Second, the heterogeneous nature of lesions (both for ICH and CI) impacts the entire cerebral system (not only specific movement impairment), which both directly and indirectly influences bilateral motor control. For example, upper-limb ataxia due to brainstem bleeding or post-stroke cognitive decline disrupts voluntary control and may provoke abnormal pain sensations (including hemiplegic shoulder pain) during movements [3,6].

Considering those limitations and other attempts in stroke hand gesture decoding [29,30,31,32,33], we instead decided to focus on extracted feature vectors precisely by combining TM4 and TM5 for both datasets, but AR3 and LPC3 for the paretic dataset only. These coeffects have a close implementation albeit focusing on different events in total (autoregressive coeffects are recurrent, whereas linear predictive coefficients rely on calculation of gain function for each order cycle). Additionally, logarithmic versions of scalable functions (LCARD and LRMSV2-3) were used to manage filtering cut-out settings and root-mean-square envelope normalization. In our case, several features were added precisely to enhance vector weights per channel and outline the static signal structure of myoelectrical input. It is worth noting that consideration of abnormal noise in the paretic extremity required different features for each dataset: AAC for non-affected and MFL for affected. Lastly, taking advantage of high sampling rates, we implemented signal transformations [53,60] and stored them as features using *Db4* for affected and *Db8* for non-affected (these settings turn out to be sensitive in total accuracy performance validation).

Over the course of the study, using supervised learning and stroke-oriented feature sets (ASF-14NP, ASF-24P) extracted from simple sEMG signal input, we were able to discriminate static hand gestures with statistically viable results (*p* < 0.05) between classifier groups. These results show that our method is suitable for practical application since we broke through the pre-determined 70% accuracy threshold for human–computer interfaces in rehabilitation [35]. Moreover, we achieved reliable myoelectrical processing prediction performance, outperforming or having comparable results to related experiments in stroke decoding [29,30]. This is likely due to root-mean-square envelope normalization, advance filtering, and a comprehensive feature set. 

In general, our results suggest that the behavior of the unaffected limb is specific but tends to be similar to those reported in healthy individuals’ gestures (especially regarding time-domain components) [33,36,38]. We revealed that, in most scenarios, SVM had the best accuracy rate scores (94.5–95%; *p* < 0.05). For SVM, PCA downgraded the prediction models for each *GL* on average by less than 2% while having only one-third the length of a concatenated feature vector. Maintenance of such a shorter length of the extracted feature vector, coupled with predefined optimal PCs values, provided competitive model accuracy to the initial feature vector. For real-time use, even with lower accuracy scores, PCA application for SVM can be beneficial since dimensionally redacted ASF feature vectors will be processed faster and, at the same time, will serve as a primitive filter (by cutting off sections of the signal input) [65]. In the non-affected dataset, the k-NN algorithm revealed specific events that were associated with stable inter-class deviation for fine movement of pinch and thumb. Conversely, the feature vector’s dimensional shifting for k-NN in general turned out to be non-sensitive. Therefore, PCA makes k-NN practical during advanced or strict filtering where only a fractional segment of the ASF feature vector is available. LDA turns out to be less efficient for the non-affected side (Table 5), having the lowest performance predictors (especially for the simplest *GL4* model), while the capabilities of the linear classifier increased with the number of gesture predictors. However, PCs increased mean accuracy for *GL*4 by 6.5% and 1–2% for the remaining *GLs*.

As for the affected side, myoelectrical decoding without using PCs revealed that only SVM had outstanding performance metric scores usable across a set of gesture models. Adding the PCA method explicitly increased LDA performance by 12–13% in some scenarios, while PCs downgraded model performance both for k-NN and SVM. However, depending on the number of predicted labels used (4–7), certain algorithms have advantages with similar accuracy rate trends in both extremities.

In contrast to our results, Zhang and Zhou [33] used high-density sEMG gesture prediction (using AR6 and RMS) in chronic post-stroke patients and revealed that PCs contrarily increased accuracy for SVM. From these and related studies [38,59], we hypothesized that the number of channels (source for feature per channel input) and higher-order parametrical functions elongate concatenated feature vectors to drastically change PCA behavior and other dimensional reduction techniques [35]. This observation should be considered for a useful approach to paretic extremity decoding.

During the experiments, we discovered that the fist (mostly mismatched with wrist flexion and thumb) is the most complicated in terms of random gesture prediction (especially for the paretic side), whereas rest state and wrist extension had the best prediction. Nonetheless, by referring to related studies conducted on stroke patients, we discovered at least several cases of similar tendencies in fist-like gesture recognition. As an example, Yang et al. (12 patients) obtained lower accuracy scores precisely in grip movements both for main and control groups among eight static hand gestures [56]. Theoretically, this phenomenon can be explained by crosstalk within the tight anatomical arrangement of forearm muscles and the low number of sEMG electrodes [18,41]. An analogous observation was illustrated by Phinyomark et al. [27] where the fist gesture in terms of signal intensity had a dominant signal input in comparison to wrist flexion and extension. Likewise, it should be noted that the complexity of the forearm and hand musculoskeletal signal evaluation is controlled by the ratio of the superficial layer of flexors and extensors under a specific electrode placement to manage the level of muscular crosstalk [18,42]. Therefore, even a two-channel system in certain scenarios can reliably classify basic hand gestures [32] while eight-channel or mesh-electrodes focus on spatial relationships of the signal structure for ANN classification [25,54,61]. For example, sEMG bracelet studies [16,27] showed that fist or other grip-related gestures tend to have higher scores. In such practices, the research design implies that sensors envelop the forearm at a single interval, and classification usually interprets myoelectrical data in accordance with inertial measurement unit (IMU) sensors. However, these bracelet-like sensors are likely to be less appropriate for paresis at the acute stage of stroke since the applicability of such devices is limited due to low sampling rates and an inability to adjust to specific muscle areas. Likewise, IMU sensors would be inefficient due to lack of spatial coordination in cases of severe paresis (which is widely encountered across the stroke demographic) [11,14]. With respect to medical applications, these trends should be considered for early post-acute stroke.

On the contrary, gesture models without the fist gesture in both datasets generally saw increases in total accuracy results by an average of 7–9% after model complexity reduction [43,48]. Within our system, electrode placement on the hand (thenar and hypothenar area in addition to the forearm) was also crucial as our initial basic setup exploited BS scores and their importance in fine hand paretic evaluation by adding a single bipolar electrode on the thenar area to translate abductor and flexor pollicis brevis activity. In future experiments, as was also suggested by Jochumsen et al. [30], we are planning to add another pair of electrodes to the hypothenar area, focusing on flexor and abductor digiti minimi that might lead to wider hand signal distribution in sEMG signal context per gesture (such as in finger spreading).

During testing of the difference between classifiers (*p* < 0.05), we discovered an interesting observation that, within each *GL*, the statistically viable pairs in both datasets had precisely specific patterns in spite of the model complexity and number of PCs (five, 19, and 23). Thus, additional insight related to paretic *GL6* and *GL7* combinations, in which significance between the groups was achieved only within a single inter-class of classifiers between the affected and non-affected sides, is required from future experiments on an expanded patient population. For this, we hypothesize that, for each stroke-oriented feature set, the classifier behavior has a specific pattern depending on dimensional reduction [56,65] which can be tuned toward the particular gesture of interest. Lastly, since PCA reduces computational effort and targets certain synergetic active myoelectrical patterns in certain applications, it was judged essential for real-time use.

We acknowledge the limitations of our study. Our results could be seen as inferior to previous studies carried out on healthy individuals or chronic stroke survivors, and they require detailed settings of technical and clinical aspects. Even though our method demonstrates practical usability for post-acute stroke survivors and reveals several trends that might serve as guidelines, the enhanced complexity could be a barrier to adoption. 

In the future, we will validate such a system on new patients (including real-time evaluation), improve the prediction capabilities, generate advances in signal pre-processing, and optimize classifiers. These novel ASF feature sets require additional attention and further testing to validate their performance parameters (e.g., frequency settings, increased number of gestures, and more trainee data from an acute stroke).

## 5. Conclusions

In a pilot study on 19 post-acute stroke patients, we were able to validate the practical possibility of recognizing simple hand gestures on both paretic and non-affected sides using only a four-channel sEMG and supervised learning. The evaluated total accuracy rates using SVM for four-, five-, six-, and seven-gesture models were 96.62%, 94.20%, 94.45%, and 95.57% for non-paretic and 90.37%, 88.48%, 88.60%, and 89.75% for paretic limbs, respectively. LDA had a competitive accuracy performance using PCA whereas k-NN turned out to be a less efficient classifier in gesture prediction.

For post-acute stroke application, we did find several points of interest with our system. Among these: early-stage pattern recognition should include up to five predictive labels only for supervised learning (i.e., four or seven gestures), while, for the follow-up recovery process, applied hand and forearm gestures should be distinct according to upper limb biomechanics with an aim to trigger specific flexor/extensor ratio bursts on the forearm muscles (i.e., wrist flexion and wrist extension). Lastly, myoelectrical prediction of fine gestures can be enhanced using dimensional reduction approaches.

## 6. Patents

The research results are protected by international law in a patent. This original manuscript was created during the patent pending process.

## Figures and Tables

**Figure 1 sensors-22-08733-f001:**
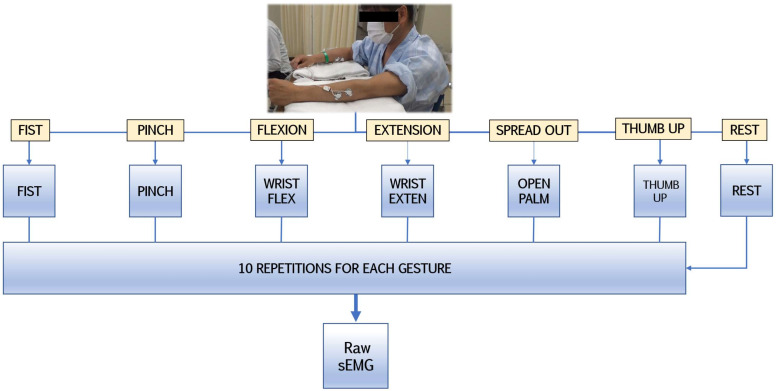
The experimental protocol for detecting sEMG signals during gestures.

**Figure 2 sensors-22-08733-f002:**
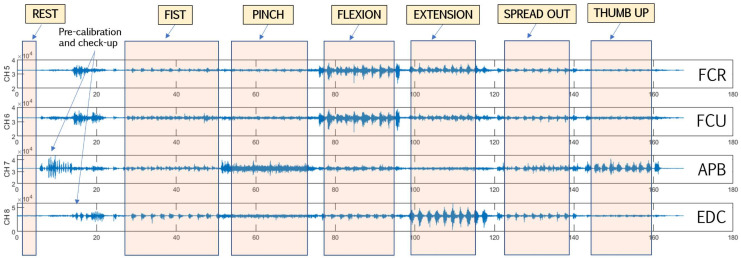
Representative four-channel sEMG data labeling and segmentation. FCR: flexor carpi radialis; FCU: flexor carpi ulnaris; APB: abductor pollicis brevis (thenar area); EDC: extensor digitorum communis.

**Figure 3 sensors-22-08733-f003:**
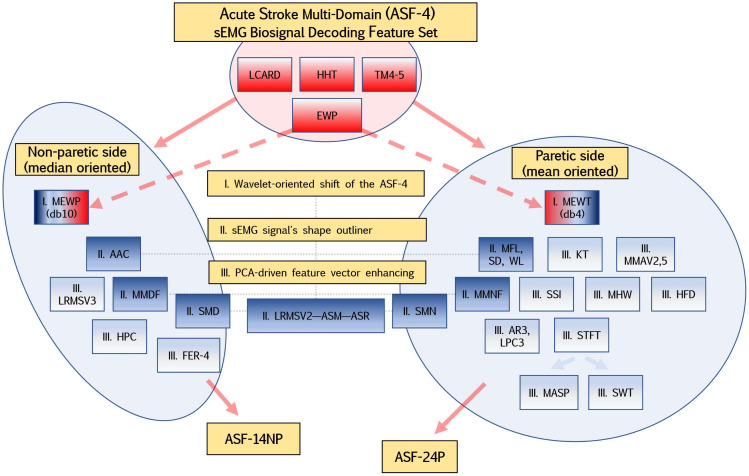
The diagram of the ASF-4 wavelet transformation and additional feature extraction into ASF-14NP and ASF-24P feature sets. The pale-red shape shows the key part of the feature sets, while the blue shapes demonstrate feature concatenation to enhance vector weight.

**Figure 4 sensors-22-08733-f004:**
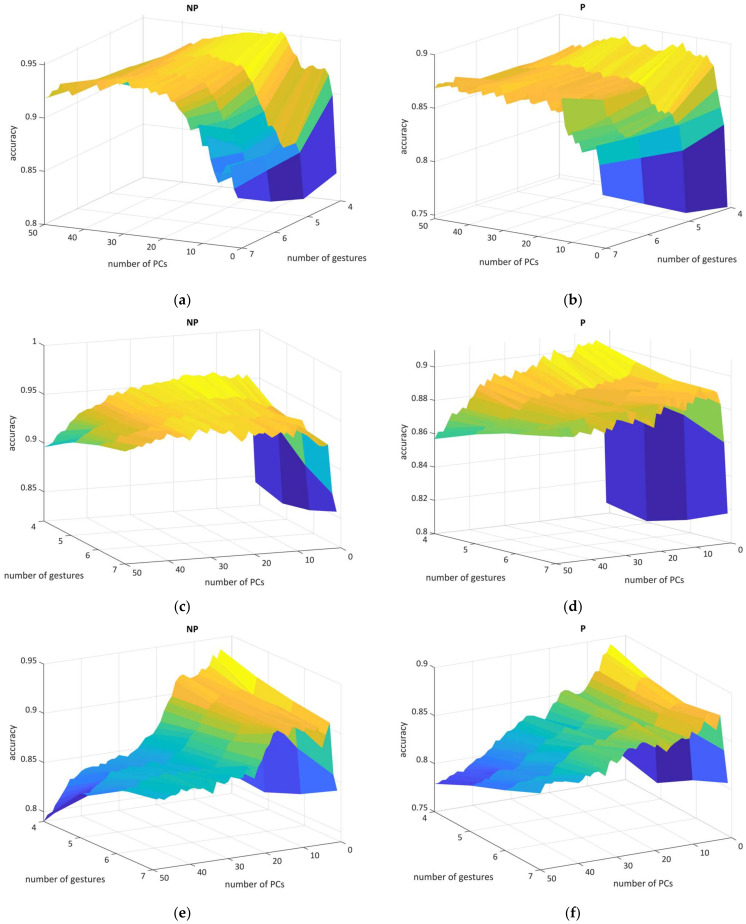
The 3D mesh illustrates the relationship between *GLs* in paretic and non-paretic datasets (left figures obtained from NP19, right from P19), number of PCs, and accuracy: (**a**,**b**) SVM, (**c**,**d**) LDA, and (**e**,**f**) k-NN performance. The color heatmap mesh represents the accuracy result range from dark blue (lower scores) to yellow (higher scores).

**Figure 5 sensors-22-08733-f005:**
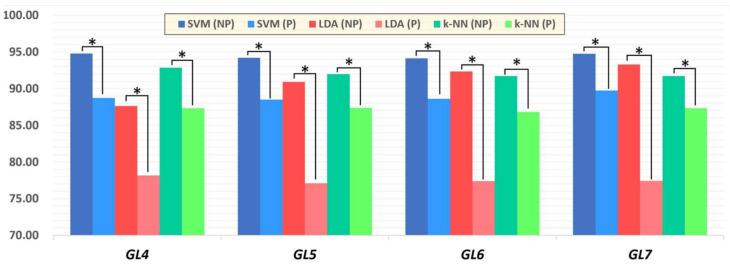
Total mean accuracy rate among *GLs* using SVM, LDA, and k-NN. Asterisks indicates statistical significance between paretic and non-paretic pairs using a paired *t*-test (* *p* < 0.05).

**Figure 6 sensors-22-08733-f006:**
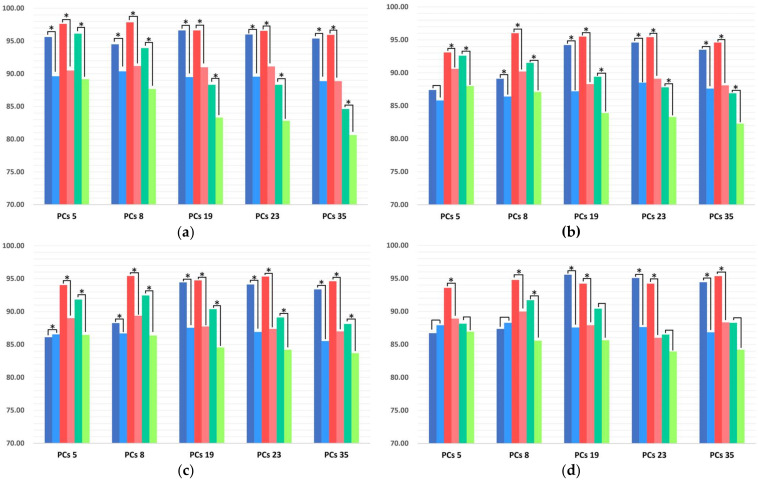
Total mean accuracy rates of *GLs* from four to seven using SVM, LDA, and k-NN, dependent on the application of PCA (* *p* < 0.05) on paretic (P19) and non-paretic (NP19) datasets: PCA-dependent *GL4* (**a**), *GL5* (**b**), *GL6* (**c**), and *GL7* (**d**) model. Legend options: dark blue (NP19) and blue (P19) bars are the *Acc* from SVM, red (NP19) and pale-red (P19) bars are the *Acc* from LDA, dark-green (NP19) and green (P19) bars are the *Acc* from k-NN. Asterisks between the paretic and non-paretic evaluations of the single classifier indicate statistical significance (* *p* < 0.05).

**Figure 7 sensors-22-08733-f007:**
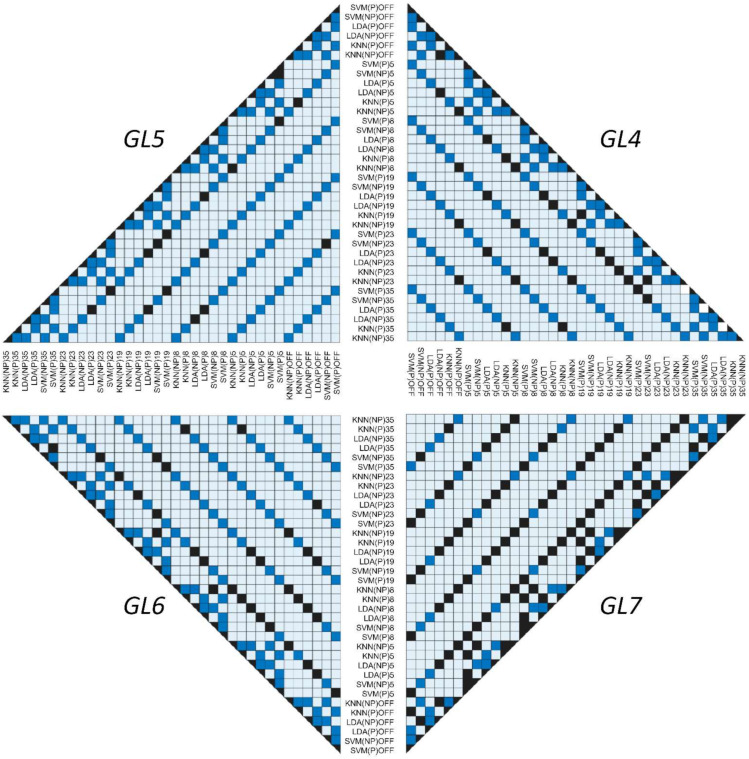
Testing difference between classifiers (SVM, LDA, and k-NN) using a paired *t*-test among paretic and non-paretic sides with and without the use of PCA: right top is *GL4*, left top is *GL5*, left bottom is *GL6*, and right bottom is *GL7* model. The dark-blue squares indicate a significance between the assessed pairs of classifiers (*p* < 0.05) and black color shows non-significance (*p* > 0.05).

**Figure 8 sensors-22-08733-f008:**
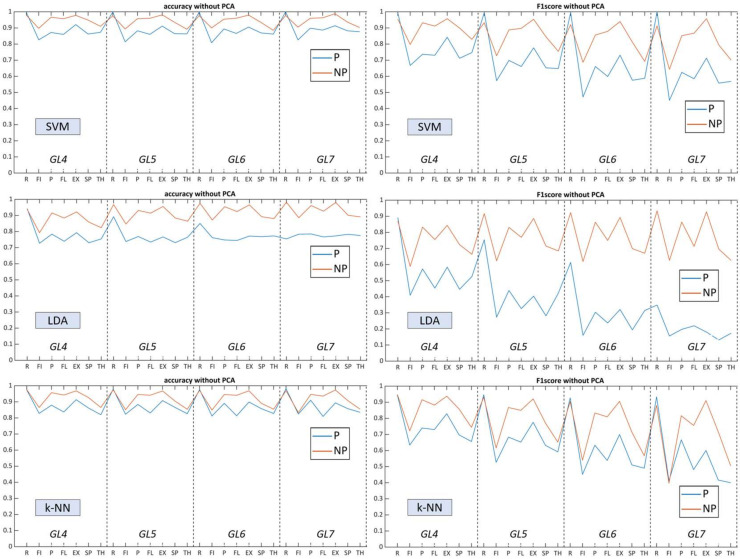
Total accuracy rate and *F*_1_-score from *GL* models on NP19 and P19 datasets without the PCA: top figures show mean *Acc* and *F*_1_ from SVM, middle figures show mean *Acc* and *F*_1_ from LDA, and bottom figures show mean *Acc* and *F*_1_ from k-NN.

**Figure 9 sensors-22-08733-f009:**
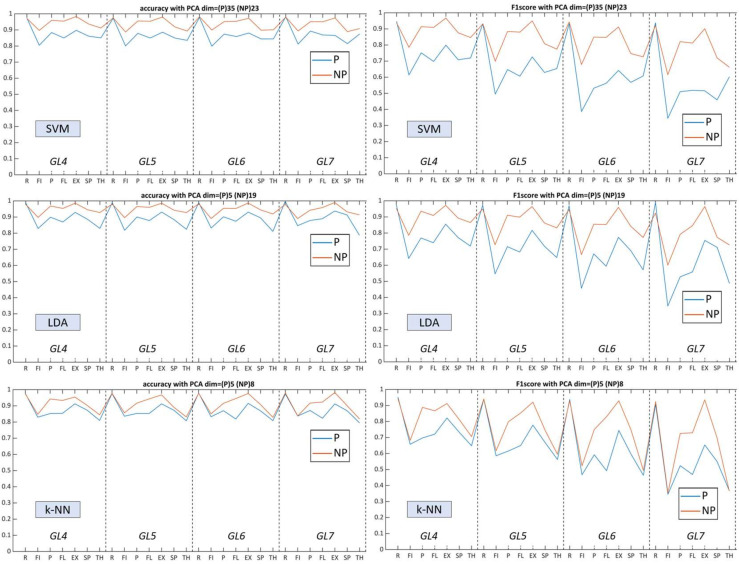
Total accuracy rate and *F*_1_-score from *GLs* models on NP19 and P19 with best PCA settings: top figures show mean *Acc* and *F*_1_ from SVM, middle figures show mean *Acc* and *F*_1_ from LDA, and bottom figures show mean *Acc* and *F*_1_ from k-NN.

**Figure 10 sensors-22-08733-f010:**
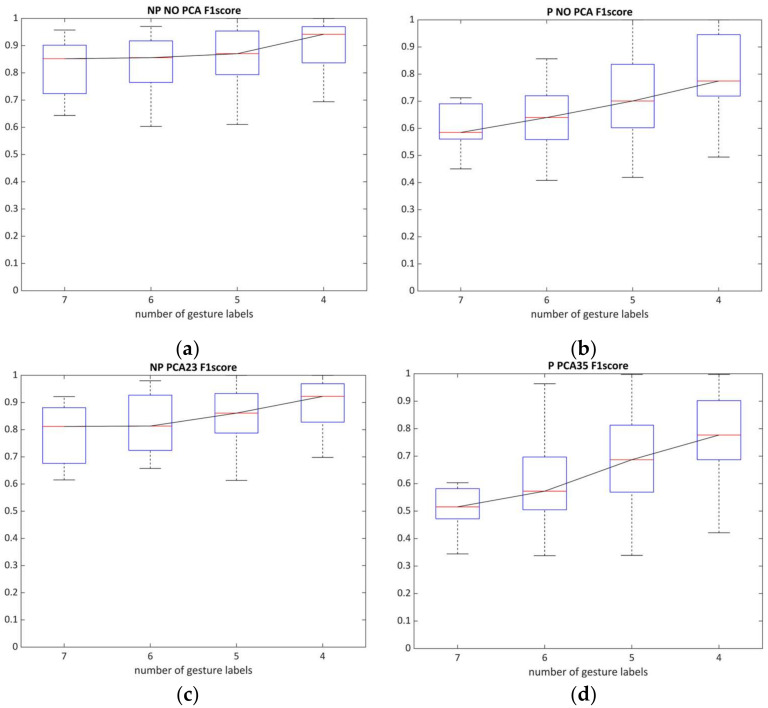
The relationship between the number of gestures and the average value of the *F*_1_-score (cross-validated by SVM with and without PCA for paretic and non-paretic datasets): (**a**,**c**) NP19 without and with the use of PCA; (**b**,**d**) P19 with and without the use of PCA.

**Table 1 sensors-22-08733-t001:** ASF-4 multidomain feature set components.

Domain	Feature	Internal Parameters	Short Description
Time domain	TM4-5	m=4, 5	$TM_{m}=\left|\frac{1}{N}\overset{N}{\underset{n-1}{\sum }}x_{n}^{m}\right|,\;$ [27].
	LCARD	Threshold set to 0.001	LCARD examines the number of unique values in the time-series set among each channel [55].
Time–frequency domain	HHT		HHT is a high-order signal processing model of empirical mode decomposition and the Hilbert transform [53,54].
	MEWP	Wavelet Daubechies (*Db4*)	MEWP provides enriched signal analysis by wavelet decomposition set of parameters: position, decomposed signal scaling, and frequency curve [43,56].

*N* is the window length, *n* is the data sample, $x_{n}$ denotes the sEMG signal in a predefined segment *n*, and *m* is the numerical number of a high-order temporal moment.

**Table 2 sensors-22-08733-t002:** Feature additions of the ASF-14NP stroke-oriented multidomain feature set.

Domain	Feature	Internal Parameters	Short Description
Time domain	LRMSV2-3	γ=2,3	$\mathrm{LRMSV}\gamma =log{\left(\frac{1}{N}\overset{N}{\underset{n=1}{\sum }}x_{n}^{\gamma }\right)}^{\frac{1}{\gamma }}$;
	ASM	$exp=\left\{\begin{array}{c} 0.5,\;if\;\left(n\geq 0.25N\;and\;n\leq 0.75\right)\\ 0.75,\;otherwise \end{array}\right.$;	$\mathrm{ASM}=\left|\frac{\sum_{n=1}^{N}{\left(x_{n}\right)}^{e}}{N}\right|$;
	ASR		$\mathrm{ASR}=\left|\overset{N}{\underset{n=1}{\sum }}{\left(x_{n}\right)}^{\frac{1}{2}}\right|$;
	AAC		$\mathrm{AAC}=\frac{1}{N}\overset{N-1}{\underset{n=1}{\sum }}\left|x_{n+1}-x_{n}\right|$;
	HPC	$Activity=\frac{1}{N}\overset{N-1}{\underset{n=0}{\sum }}{\left(x_{n}-\bar{x}\right)}^{2}$$,\;Mobility=\sqrt{\frac{Activity\left(dn/dt\right)}{Activity}}$;	$\mathrm{HPC}=\sqrt{\frac{Mobility\left(dn∕dt\right)}{Mobility}}$;
Frequency domain	MMDF		$\mathrm{MMDF}=\frac{1}{2}\overset{M}{\underset{j=1}{\sum }}A_{j}$;
	SMD		$\mathrm{SMD}=\frac{1}{2}\overset{M}{\underset{n=1}{\sum }}PSD_{n}$;
Spatial domain	FER-4	The normalized mean value of the ratio between flexors and extensors channels.	

$A_{j}$ represents the signal’s amplitude spectrum in a certain bin socket *j, M* is the frequency bin length (MMDF), and *PSD* is power spectral density of the signal (*SMD*).

**Table 3 sensors-22-08733-t003:** Feature additions of the ASF-24P stroke-oriented multidomain feature set.

Domain	Feature	Internal Parameters	Short Description
Time domain	MMAV2,MMAV5	$w_{n}=\left\{\begin{array}{c} 1,\;0.25N\leq n\leq \;0.75N\\ \frac{4n}{N},\;n<0.25N\\ \frac{4\left(N-n\right)}{N},\;otherwise \end{array}\right.$; $w_{n}=\left\{\begin{array}{c} \;\frac{4n}{N},\;n<0.25N\;\\ \frac{4n}{N}-1,\;0.25N\leq n\leq 0.5N\\ \frac{4n}{N}-2,\;0.5N\leq n\leq \;0.75N\\ \frac{4n}{N}-3,\;otherwise \end{array}\right.$;	$\mathrm{MMAV}2=\frac{1}{N}\overset{N}{\underset{n=1}{\sum }}w_{n}\left|x_{n}\right|$; $\mathrm{MMAV}5\;\mathrm{splits}\;\mathrm{the}\;\mathrm{signal}\text{’}s\;\mathrm{energy}\;\mathrm{window}\;(w_{n}$) of interest aiming to investigate the 3/5th of 4/5th window segment. Equitation is similar to *MMAV2* [25,57].
	SSI		$\mathrm{SSI}=\overset{N}{\underset{n=1}{\sum }}{\left|x_{n}\right|}^{2}$;
	KURT		$\mathrm{KT}=\frac{N\left(N+1\right)}{\left(N-1\right)\left(N-2\right)\left(N-3\right)}\overset{N}{\underset{n=1}{\sum }}{\left(x_{n}-\bar{x}\right)}^{4}-3\frac{{\left(N-1\right)}^{2}}{\left(N-2\right)\left(N-3\right)}$;
	SD		$\mathrm{SD}=\sqrt{\frac{1}{N}\underset{n=1}{\sum }{\left(x_{n}-\bar{x}\right)}^{2}}$;
	MFL		$\mathrm{MFL}={log}_{10}\left(\sqrt{\overset{N}{\underset{n=1}{\sum }}{\left(x_{n}-x_{n+1}\right)}^{2}}\right)$;
	WL	Threshold set to 0.05	$\mathrm{WL}=\overset{N-1}{\underset{n=1}{\sum }}\left|x_{n+1}-x_{n}\right|$;
	MHW		$\mathrm{MHW}=\overset{N}{\underset{n=0}{\sum }}{\left(W_{n}x_{n}\right)}^{2}$;
	AR3	$x_{n}=a_{0}+ar_{1}\left(x_{n}-1\right)+ar_{2}\left(x_{n}-2\right)+ar_{3}\left(x_{n}-3\right)$;	$\mathrm{AR}3=\left[a_{0},ar_{1},ar_{2},ar_{3}\right]$, order set to 3
	LPC3	$x_{n}=b_{0}+b_{1}x_{n}+b_{2}(x_{n}-1)+b_{3}\left(x_{n}-2\right)$;	$\mathrm{LPC}3=\left[b_{0},b_{1},b_{2},b_{3}\right]$, order set to 3
Frequency domain	MASP	$\mathrm{Fast}\;\mathrm{Fourier}\;\mathrm{transform}\;\mathrm{parameters}\;\mathrm{are}\;\mathrm{split}\;\mathrm{into}\;5\;\mathrm{bins}\;fft_{k}\left(k=1,\;2,\;3,\;4,\;5\right)$	$\mathrm{MASP}=\overset{N}{\underset{n=1}{\sum }}\frac{\left|fft_{k}\right|}{N}$;
	SMN		$\mathrm{SMN}=\frac{\sum_{j=1}^{M}f_{j}PSD_{j}}{\sum_{j=1}^{M}PSD_{j}}$;
	*MMNF*		$\mathrm{MMNF}=\frac{\sum_{j=1}^{M}f_{j}A_{j}}{\sum_{j=1}^{M}A_{j}}$;
Time–frequency domain	*STFT*		$\mathrm{SFTT}=\overset{N-1}{\underset{r=1}{\sum }}x\left(r\right)g\left(r-n\right)\epsilon^{-j2\pi mi/N}$;
	*EWT*	Wavelet Daubechies (*Db4*)	$\mathrm{EWT}=\sqrt{\frac{1}{K}\overset{K}{\underset{k=1}{\sum }}W_{J,k}^{2}}$;
	STW	STW is a noise-resilience method of wavelet transform and STFT to highlight signal window length other than artefacts or defect stochastic window frames [60].	
Fractal domain	HFD	HFD evaluates muscle strength and the contraction grade; it measures the size and complexity of the sEMG signal in the time-domain spectrum without fractal attractor reconstruction methods [58].	

*g* is a window function (*STFT*); $f_{j}$ is a frequency of the signal amplitude spectrum (*MMNF*, *SMN*); $W_{n}$ is a Hamming window function (*MHW*); *K* is the number of the *j*-th layer decomposed coefficient level, and $W_{J,k}$ is the *k*-th coefficient of the given layer decomposed coefficients (*EWT*).

**Table 4 sensors-22-08733-t004:** Post-stroke patient demographics and BS, SIAS, and FMA-UE scores.

Patient	Age, Gender	Lesion	Days Since Onset	BS	SIAS	FMA-UE	MAS	Affected Side
HGR-001	80, M	CI	11	5, 5	4, 4	58	0, 0, 0	R
HGR-002	32, F	CI	11	5, 5	4, 4	50	0, 0, 0	R
HGR-003	71, M	ICH	13	5, 5	4, 4	64	1, 0, 0	R
HGR-004	52, F	CI	8	6, 6	5, 5	63	0, 0, 0	L
HGR-005	82, M	CI	9	4, 3	3, 1	27	1, 0, 0	L
HGR-006	81, M	ICH	5	2, 4	1, 1	16	0, 0, 0	R
HGR-007	77, M	CI	9	6, 6	5, 5	60	0, 0, 0	R
HGR-008	79, F	ICH	7	3, 4	2, 3	28	0, 1+, 1+	R
HGR-009	65, M	ICH	5	6, 6	5, 5	55	0, 0, 0	R
HGR-010	67, F	ICH	12	5, 4	3, 1	37	0, 0, 0	L
HGR-011	66, M	CI	13	3, 3	2, 1	15	1+, 0, 0	L
HGR-012	64, F	ICH	33	4, 5	3, 4	35	1, 1, 0	L
HGR-013	63, M	CI	13	5, 5	4, 4	59	1, 0, 0	R
HGR-014	50, M	CI	19	3, 3	2, 1	22	1, 1, 0	R
HGR-015	72, F	CI	16	6, 5	5, 4	52	0, 0, 0	R
HGR-016	57, M	ICH	12	5, 4	4, 4	42	0, 0, 0	L
HGR-017	57, M	ICH	18	2, 2	1, 0	8	0, 0, 0	L
HGR-018	64, M	CI	9	6, 6	4, 4	60	0, 0, 0	R
HGR-019	74, F	CI	12	2, 1	1, 0	9	0, 1, 0	L

‘+’ in the MAS column indicates spasticity progression. MAS evaluates only the affected upper extremities; it has triple parameters which represent the spasticity of the elbow, wrist, and fingers. BS and SIAS have binary parameters which represent the forearm and hand.

**Table 5 sensors-22-08733-t005:** Mean average value of accuracy and *F*_1_ scores of supervised gesture recognition from four to seven gesture labels (based on the hand movements of 19 stroke patients) without and with PCA.

	Gesture Classification without PCA (*GL4*, *GL5*, *GL6*, *GL7*)				Gesture Classification with PCA * (*GL4*, *GL5*, *GL6*, *GL7*)			
	Non-Paretic Side (NP19)		Paretic Side (P19)		Non-Paretic Side (NP19)		Paretic Side (P19)	
Classifier	*Acc* (%)	*F*_1_ (%)	*Acc* (%)	*F*_1_ (%)	*Acc* (%)	*F*_1_ (%)	*Acc* (%)	*F*_1_ (%)
SVM	94.80	89.74	88.71	77.53	92.81	85.63	87.59	74.90
	94.19	85.69	88.48	71.49	91.41	78.77	86.07	65.16
	94.13	82.67	88.60	66.01	91.36	74.33	85.98	57.82
	94.73	81.82	89.75	64.26	91.97	72.00	86.40	52.20
LDA	87.63	75.42	78.16	55.48	94.05	88.13	88.64	77.15
	90.92	77.52	77.09	41.42	93.79	84.55	87.96	69.94
	92.35	77.39	77.39	30.63	93.85	81.66	87.97	63.98
	93.27	76.95	77.43	20.09	94.02	79.25	88.17	58.61
k-NN	92.84	85.82	87.32	74.73	86.30	72.58	82.64	65.36
	91.98	80.10	87.38	68.67	87.12	68.03	83.26	58.37
	91.70	75.34	86.81	60.72	87.70	63.76	84.24	53.03
	91.71	71.12	87.31	55.86	88.19	59.61	85.23	48.49

* PCA had the best specific parameters for each classifier and extremity side; *GL* indicates the mean scores of all possible gesture combinations, while integer values of 4–7 specify the model complexity (number of gesture labels).

## Data Availability

Data obtained from the research are not subject to public access in accordance with Japanese law and the ruling of the Clinical Ethics Review Board. For more detailed comments and requests please contact H.K. and A.M. (Aiki Marushima): kadone@ccr.tsukuba.ac.jp and aiki.marushima@md.tsukuba.ac.jp.

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
