# Peer review of "Supervised Myoelectrical Hand Gesture Recognition in Post-Acute Stroke Patients with Upper Limb Paresis on Affected and Non-Affected Sides"

_sensors, 2022, doi:10.3390/s22228733_

Round 1
Reviewer 1 Report
It is good to see that the author made many contributions in conducting the data collection, data analysis, and experiment explorations. However, there are a few comments to share with the author.
Comments:
1. In the Introduction section, page 3, line 123, the author mentioned that this research of study has the benefits to “achieve user-independent application for further fine motor recovery and timely recovery monitoring diagnostics in the critical post-stroke time period”. The significance of it should be more specific, like what kinds of specific applications, why it is important. More references would be good to support such viewpoint.
2. In page 4, line 16; page5, line 196, citation issues to fix.
3. In page 4, why the seven gestures are confirmed to be the final gestures to investigate? Any medical value related. It would be good that the author could provide more reference or explanation for deciding these hand gestures for the experiment setup.
4. Please reorganize the tables to make them fit for the format.
5. In the part of machine learning strategies, why LDA, SVM, KNN are selected instead of any other prevalent machine learning techniques.
6. Figure 13 to 15 are not readable and hard to understand. It would be good to find a better way to present your results clearly. Also, figure 5 to 7, they looks repeated figures. It would be good that the authors can find another data presenting way to better reflect their experiment results.
7. The authors did many experiments’ combinations with many test variables (PCA, GL4 to GL7). They used so many figures and tables to present the experiments’ analysis, but I cannot find the significance of each identical designed setup. It would be good that the author can provide more explanations or evidences to the experiments’ setups.
8. The discussion section is too long to capture the main viewpoints.
9. The conclusion is too vague to clarify some key identifications. The conclusion part only listed some accuracies of the classifiers. It would be good that the authors can find some interesting outcomes to discuss or conclude some key points via the experiments.
Reviewer 2 Report
I think the contribution of the article is good, the fact of forming a set of features with several domains is something new, however, there are many things that are not justified or poorly explained. The article is very difficult to read. So, the main suggestion is to review the writing, to improve it and correct grammatical and spelling errors. In addition, to reviewing the citations, there are many things wrong and contemplating the idea of restructuring some sections.
The specific comments and/or doubts are the following. If pertinent, the answers must be included in the article:
1. Line 26
You can justify the use of only four channels with “A Study of Movement Classification of the Lower Limb Based on up to 4-EMG Channels”.
2. Line 199
Why did the Butterworth filter go from 20Hz to 300Hz?
3. Line 200
Why more than double the STD?
4. Lines 209-213
Why were those decisions made?
5. Line 214
Why do those limits make it a "valid" standardization?
6. Lines 217-221
How was the sEMG data inspected to avoid maximum misdetection?
7. What does the unlabeled segment in Figure 2 mean?
8. Figure 3 is not a flowchart.
9. Where is Table 1 cited in line 252?
10. In tables 2 and 3 some variables are not explained.
11. Lines 329 – 331
What is visualization for? What is the contribution?
12. Figure 4
What does the “*” mean?
It is not clear how the precision was obtained.
13. Tables 5 and 6 could be summarize in text and they could be removed from the article.
14. The content of Table 7 does not correspond to what is mentioned in your citation on line 447, and Table 8 does not exist.
15. Figures 5, 6 and 7 are not understood, they look very similar, besides their contribution is not understood, so I suggest that they be removed from the article.
16. There is no discussion about Figure 18.
17. Line 664 It is mentioned that the hypothesis was validated, but it has not been said exactly what it is, it is convenient to add it.
18. The titles of the figures and tables are over-explained, it is not necessary to explain again the acronyms that have already been explained in the text.
19. The histogram is used to visualize data distribution set, so the figures aren’t histograms, they are simply bar graphs.
20. There are a lot of numbers on the graphs that don't look good.
Round 2
Reviewer 1 Report
Thanks for making the revision.
I think most of the issues are appropriately addressed by the authors.
Author Response
Response to the Reviewer 1. Round 2.
Thanks for making the revision. I think most of the issues are appropriately addressed by the authors.
Response: Thank you very much for the positive comments and the review, which helped to improve our manuscript significantly. The article has been proofread by the native English language editor. We will follow the suggestions regarding the style and design also in our next study.
Reviewer 2 Report
- Line 625, "et al." should be in italic.
- Some words are badly cut off at the end of the line.
- Please check your references, the order in which they are cited isn't correct. There are even some that aren't mentioned.
Author Response
Response to the Reviewer 2. Round 2.
Response: Thank you very much for the review and the comments. We greatly appreciate your work regarding the evaluation of our manuscript.
Comments:
- Line 625, "et al." should be in italic.
Response: Yes, we have corrected it as suggested.
- Some words are badly cut off at the end of the line.
Response: We sincerely thank you for this valuable comment. While editing the links to the sources, we partially tried to correct the line boundaries. However, the main sections of the manuscript (due to large images) are fixed on a certain number of pages, and interference with the text rows causes the pages to shift to the next page, making it more difficult to read. We will fix this aspect together with the editorial process to improve the quality of the manuscript presentation.
- Please check your references, the order in which they are cited isn't correct. There are even some that aren't mentioned.
Response: We are grateful for the indication regarding references in our hand gesture recognition study. Indeed, in our manuscript, we referred to the clinical and technical aspects of machine learning and stroke, which was reflected in a large number of references. We carefully revised the list, reduced the volume, and reorganized that part of the manuscript. Almost all the references have been renumbered.